# Urinary Biomarkers for Early Diagnosis of Lung Cancer

**DOI:** 10.3390/jcm10081723

**Published:** 2021-04-16

**Authors:** Roberto Gasparri, Giulia Sedda, Valentina Caminiti, Patrick Maisonneuve, Elena Prisciandaro, Lorenzo Spaggiari

**Affiliations:** 1Department of Thoracic Surgery, IEO, European Institute of Oncology IRCCS, Via Giuseppe Ripamonti 435, 20141 Milan, Italy; giulia.sedda@ieo.it (G.S.); valentina.caminiti@ieo.it (V.C.); elena.prisciandaro@ieo.it (E.P.); lorenzo.spaggiari@ieo.it (L.S.); 2Division of Epidemiology and Biostatistics, IEO, European Institute of Oncology IRCCS, Via Giuseppe Ripamonti 435, 20141 Milan, Italy; patrick.maisonneuve@ieo.it; 3Department of Oncology and Hemato-Oncology, University of Milan, Via Festa del Perdono 7, 20122 Milan, Italy

**Keywords:** lung cancer, urine, biomarker, early diagnosis

## Abstract

Lung cancer is the leading cause of cancer deaths worldwide. Its early detection has the potential to significantly impact the burden of the disease. The screening and diagnostic techniques in current use suffer from limited specificity. The need therefore arises for a reliable biomarker to identify the disease earlier, which can be integrated into a test. This test would also allow for the recurrence risk after surgery to be stratified. In this context, urine could represent a non-invasive alternative matrix, with the urinary metabolomic profile offering a potential source for the discovery of diagnostic biomarkers. This paper aims to examine the current state of research and the potential for translation into clinical practice.

## 1. Introduction

Lung cancer has a high mortality rate globally, and in the majority of cases, diagnosis is often made at a late stage when the process of metastatization has already begun [1]. Thus, patient survival has been limited over the last 20 years [2]. This unfavorable outcome is mainly due to the absence of an easy-to-perform, accurate, non-invasive diagnostic test for the population at risk.

So far, in clinical practice, low-dose computed tomography (LDCT) is the only screening test validated for the early diagnosis of lung cancer in symptomatic subjects or screening of selected risk categories, such as heavy smokers over 50 years of age [3,4]. It has been demonstrated that for well-selected high-risk subjects, LDCT can promote a 20–39% reduction in the number of deaths due to lung cancer compared to chest X-ray or non-intervention procedures [5]. However, this does not apply to the entire high-risk population due to method costs, over-diagnosis, and the increased rate of false-positive results (approximately one in five LDCT screenings) that can lead to stressful experiences or more invasive tests [1].

Alongside increased understanding of the interactions between metabolism and cancer biology [6] associated with technological improvement, new methods have been developed and integrated into clinical practice, such as liquid biopsy [7]. For instance, in advanced lung cancer patients, liquid biopsies allow investigators to detect the presence of a single mutation or panel of mutations (e.g., EGFR or ALK mutations), enabling a personalized therapeutic strategy to be implemented for each patient [8,9]. Considering the heterogeneity of lung cancer [10,11], the need to integrate not only mutational analysis but also all the clinical features of each patient into a major complex algorithm is an issue that has been raised [12]. Investigators agree that the analysis should embrace the entirety of the tumor profile, suggesting an integration of the different levels of analysis, and the personal epidemiological data of the individual. Based on biological fluid analysis, omics researchers have consequently implemented pilot studies that have revealed great potential for early lung cancer diagnosis [13,14].

Among the different biological fluids (such as breath, blood, and bronchoalveolar fluid), urine offers several advantages: it can be collected from large cohorts, its collection is non-invasive, it incurs low handling costs, and prolonged frozen storage is possible. Moreover, the technology [15] has now evolved to the point whereby urine analysis may be performed as a multilevel approach.

## 2. The Role of Kidney Physiology in Oncological Practice

The kidneys are responsible for the elimination of endogenous compounds, drugs, and nondrug xenobiotics. Renal clearance is normally considered the net result of glomerular filtration, tubular secretion, and reabsorption. Characterization of the contribution of individual transporters expressed on basolateral and apical membranes of the tubule epithelium to drug and chemical excretion has advanced significantly over the last two decades [16].

Urine, through renal filtration, is formed by all the metabolites produced by physiological cellular catabolism. Renal filtration results in a matrix that is less complex, and results in the presence of fewer factors known to interfere with biomarker assay [17]. For many years, it has been understood that urine is composed of glucose, ketones, and metabolite products. The majority of the proteins are instead reabsorbed at the glomerular level; thus, the urine proteome is considered to be less complex than the plasma proteome. Furthermore, the presence of high-abundance proteins (e.g., albumin, alpha-1-antitrypsin, immunoglobulin) that mask low-abundance proteins have not yet allowed the whole proteome to be classified precisely [17].

Finally, cancer interaction with the cellular host on several metabolic and biological mechanisms can induce the liberation of specific cancer-correlated metabolites.

Thanks to all these characteristics, researchers have centered their efforts on finding urinary metabolites for several cancers, such as those of the urological system (kidney, prostate, and bladder) and also those of the breast, ovary, and gastrointestinal tract [18,19].

Furthermore, recent technological advances in instrumentation and equipment, such as that used in nuclear magnetic resonance, mass spectrometry, and gas and liquid chromatography, have increased the chances of discovering urine onco-biomarkers and of analytical reproducibility [19].

## 3. Materials and Methods

We carried out a comprehensive literature search using PubMed to retrieve original research papers presenting data on urinary biomarkers for the early diagnosis of lung cancer.

No language restrictions were applied, and the date restriction was from the last ten years (2010–2020) to consider the improvement in technology. Putative biomarkers were evaluated based on the five-phase approach [17] to guarantee a scientific standard as well as a roadmap for successfully translating biomarker research from the basic science to the bedside. The following PubMed search query was used as a first step: (“urine” [All Fields] AND (“biomarker s” [All Fields] OR “biomarkers” [MeSH Terms] OR “biomarkers” [All Fields] OR “biomarker” [All Fields] OR “metabolite” [All Fields] OR “metabolites” [All Fields]) AND (“lung cancer” [All Fields] OR “lung neoplasms” [MeSH Terms]) AND (“2010” [Date-Entry]: “2020/10” [Date-Entry])). Titles and abstracts available in PubMed of all identified articles were screened to ascertain their relevance. The full texts of potentially relevant study reports were further evaluated. Additional study reports identified from other sources (Web of Science, Google scholar, Embase, Scopus, and the Cochrane Library, as well as citations in the reference lists of identified relevant articles or reviews on the topic) were also evaluated for inclusion. Selected articles were reviewed and data on the diagnostic performance (sensitivity, specificity, accuracy) of various urine metabolites for the detection of lung cancer were extracted and crosschecked independently by two investigators (RG and PM). Any disagreement was resolved by their joint consensus. Similarly, we carried out a second literature search using the following PubMed search query: (“urine” [All Fields] AND (“dog” [All Fields] OR “dogs” [All Fields]) AND (“lung cancer” [All Fields] OR “lung neoplasms” [MeSH Terms]) AND (“2010/01” [Date-Entry]: “2020/10” [Date-Entry]), to retrieve studies assessing lung cancer detection by sniffer dogs.

Overall, 263 references published from 1 January 2010 to 31 October 2020 were retrieved from the first PubMed query. After excluding irrelevant papers (reviews, animal or fundamental research studies), 20 articles satisfied the selection criteria and were included in the review. A single report identified using another approach was also included. Finally, six articles were retrieved from the second PubMed query and two studies on sniffer dogs were included in the review.

## 4. Results

From the papers extracted, two studies [20,21] reported on the training of sniffer dogs. Detection dogs are currently used to identify illegal substances, such as explosives or drugs, or to recognize missing persons in highly demanding environments. These recently published studies reported the ability of trained dogs to differentiate cancer patients from healthy individuals based on urine sniffing (Table 1). These results indicate that there are determinant molecules in the urine, predictive of lung cancer.

With these data as a starting point, in recent decades, human translational approaches have been developed. Many groups have analyzed urinary metabolites employing either gas or liquid chromatography coupled with mass spectrometry to make an early screening diagnosis or detect lung cancer recurrence (Table 2).

The most extensive study produced by the National Cancer Institute, published in 2014, was qualitative with a well-established design. They used mass spectrometry in a case-control study to assess the urine of over 1000 samples and uncovered a set of urine metabolites associated with a cancer diagnosis. Two metabolites, creatine riboside and N-acetylneuraminic acid, were significantly elevated in lung cancer patients. These results were subsequently validated in an independent sample set. Both metabolites were enriched in tumor tissue compared with adjacent non-tumor tissue and positively correlated with urine levels, thus revealing their direct association with tumor metabolism [22].

A successive evaluation of this panel of urinary metabolite lung cancer biomarkers in the well-characterized prospective Southern Community Cohort Study (SCCS) confirmed the association of creatine riboside and N-acetylneuraminic acid levels with lung cancer risk before the onset of clinically-detectable disease [24].

Seow, in 2019 [23], in a nested case-control study of 564 never-smoking women, found that 5-methyl-2-furoic acid in urine was associated with a decreased risk of lung cancer.

Finally, in 2020, Patel [26] improved the detection and precise quantification of the urinary cancer metabolite biomarkers creatine riboside and creatinine riboside, creatine and creatinine, analyzing 76 lung cancer patients and 98 controls, by precise ultra-pressure liquid chromatography-tandem mass spectrometry.

Moreover, several signatures have been investigated with promising results. Zhang’s group [28] selected a panel of five urinary molecules (ferritin light chain, mitogen-Activated Protein Kinase 1 Interacting Protein 1-Like, fibrinogen Beta Chain, two Member RAS Oncogene Family, RAB33B and RAB15) as a predictive model to differentiate lung cancer from healthy lung tissue. Carrola et al. [27] instead, reported their signature with hydroxyisovalerate, R-hydroxyisobutyrate, N-acetylglutamine, and creatinine in 125 individuals, with a good performance of sensitivity and specificity, including a single molecule, such as creatine.

Yuan, in 2014 [25], assessed the lung cancer risk via urinary constituents deriving from tobacco smoke, demonstrated a significantly different risk of lung cancer according to ethnic/racial characteristics.

Additional studies have been conducted using alternative techniques, such as ELISA and PCR (Table 3) to analyze urinary metabolites.

The colloid gold aggregation procedure was employed by Takahashi’s group [31,32] in two consecutive publications, which identified diacetylspermine as biomarkers in lung cancer patients.

In 2015, Mazzone [33] analyzed the volatile organic compounds (VOCs) of the urinary headspace, finding a signature that could distinguish lung cancer patients utilizing a colorimetric sensor array exposed to the headspace gas of neat and pre-treated urine cancerous samples. Other studies [34,35,36,37] used immunosorbent assays, such as enzyme-linked immunosorbent assay (ELISA).

Finally, two interesting studies explored the genome and epigenome level. In 2019 [40], Wu and colleagues published a prospective study detecting comparable profiles of cell-free DNA in sputum, plasma, urine, and tumor tissue from 50 lung cancer patients by next-generation sequencing. Liu B et al. [38] evaluated the simultaneous positive methylation of genes (*CDO1*, *TAC1*, *HOXA7*, *HOXA9*, *SOX17*, and *ZFP42*) not only in urine but also in plasma samples, suggesting putative epigenetic biomarkers.

## 5. Study Limitations

These studies underlined the potentiality of biomarkers to differentiate lung cancer patients from healthy subjects with a non-invasive, patient-friendly fluid collection. Current limitations, for example in [24,25,28], were the small sample size investigated and the absence of a standard approach. Furthermore, other investigators [22,23,26] reported that they could not adjust and control for dietary and drug intake, thus representing the inability to control for exogenous effects on metabolism. Moreover, these studies demonstrated a degree of bias, ranging from patient selection to low accuracy, and, therefore, limited their clinical translation.

## 6. Future Perspectives

The major limitation encountered so far stems from the small sample size. This will need to be remedied in future studies. Further development and validation by means of independent, routine techniques that are more operationally feasible, such as ELISA and PCR, also seem indispensable steps for future clinical development.

The discovery and validation of biomarkers calls for the implementation of a worldwide network of research centers with constant data sharing and dissemination of results. Such connections could focus on those biomarkers that are more appropriate for clinical use [42].

For this reason, the creation of a consortium would be desirable, in which biomarkers for mass population screening are discussed and evaluated.

Each contributor will need to use a variety of data-gathering approaches and methods and render all the data transparently available in the public domain. This will be translated into a worldwide big data database which could be interrogated and analyzed [43,44].

Artificial intelligence analysis will be utilized to process, overlap, and integrate the molecular biomarkers as well as clinical and epidemiological data. The results obtained will be further processed by machine learning algorithms, enabling multiple diagnostic algorithms to be created for the early diagnosis of lung cancer [45].

It should also be noted that this network will allow for standardizing methods, which are pivotal to guarantee a high level of accuracy.

Moreover, the same approach could be applied to other biological fluids, such as blood and exhaled breath, to establish and integrate profiling and discover each individual’s phenotype in the large and heterogeneous cohort of the at-risk population.

## 7. Summary

-Urine is an appealing biological fluid in terms of ease and safety of collection, and quantity.-Renal filtration also results in a less complex matrix than that of blood, containing fewer factors known to interfere with biomarker assays.-So far, many urinary metabolites have been processed. However, they await validation.-Analytical methods have been reported for the detection of urinary biomarkers.-Technological strides in urine analytical methodology have resulted in enormous progress for basic research.-These methods could be standardized and integrated into a procedure for targeted metabolomics by clinical investigators. The resulting quantification of biomarkers would offer a formidable diagnostic tool for early-stage lung cancer.

## 8. Conclusions

Currently, there are no clinically available validated urinary biomarkers for the early diagnosis of lung cancer. However, urine has been the focus of many promising research projects over the past decade.

From a research project design perspective, for the vast majority of cases, we have created a customization of research based on the tumor’s objective characteristics.

The sample size investigated, and the lack of a standard approach, limit scientific robustness. However, all these studies are of immense value in that they are paving the way towards an international repository of high-quality data and datasets that can be interrogated and analyzed globally for further investigations.

All these results indicate the many steps that have been taken in the investigation into urinary biomarkers. Of note among these are the National Cancer Institute (NCI) studies on different urinary biomarkers based on the biomarkers’ validation criteria.

Nowadays, lung cancer is considered a disease with systemic influences, in which many pathological processes interact and develop. With the necessary resources, information, tools, and unstinting dedication, this will allow for increasingly early discovery and a better chance of healing in the near future. This prospective perception includes combining and comparing the markers examined by different groups employing a worldwide big-data database.

## Figures and Tables

**Table 1 jcm-10-01723-t001:** Studies using sniffer dog detection of urinary VOCs.

Study	Population	Main Results
Amundsen T. 2014 [20]	Lung cancer (77)	Sensitivity: 60% Specificity: 29.2%
Mazzola S.M. 2020 [21]	Lung cancer (140), Controls (194)	Sensitivity: 45–73% Specificity: 89–91%

**Table 2 jcm-10-01723-t002:** Summary of the studies using gas or liquid mass spectrometry for urine metabolite analysis

Study	Population	Lung Cancer Patients (*n*)	Method	Metabolites	Main Results
Mathé E.A. 2014 [22]	1005	469	LC-MS/MS	N-acetylneuraminic acid Cortisol sulfate Creatine Riboside 561+	Accuracy = 78.1%
Seow W.J. 2019 [23]	564	275	LC-MS/MS	5-methyl-2-furoic-acid	N.R.
Haznadar M. 2016 [24]	529	178	LC-MS/MS	Creatine riboside N-acetylneuraminic acid Cortisol sulfate 561+	Sensitivity = 50% Specificity = 86%
Yuan J.M. 2014 [25]	165	82	LC-MS/MS	PheT 3-OH-Phe total OH-Phe	
Patel D.P. 2020 [26]	174	76	UPLC-ESI-MS	Creatine ribosi de Creatinine riboside Creatine Creatinine	
Carrola J. 2011 [27]	125	71	HR-NMR	hydroxyisovalerate R-hydroxyisobutyrate N-acetylglutamine Creatinine	Sensitivity = 93% Specificity = 94%
Zhang C. 2018 [28]	231	33	LC-MS/MS	FTL MAPK1IP1L FGB RAB33B RAB15	Sensitivity = 90–96.9% Specificity = 54.5–90%
Hanai Y. 2012 [29]	40	20	GC-TOF MS	2-pentanone	Sensitivity = 85–95% Specificity = 70–100%
Anton A.P. 2016 [30]	20	6	HS-PTV-MS	2-Butanone 2-Pentanone Pyrrole 2-Heptanone 2-Ethyl-1-hexanol	Sensitivity = 40–100% Specificity = 100%

LC: liquid chromatography; MS: mass spectrometry; UPLC-ESI: liquid chromatography electrospray; HR-NMR: high-resolution nuclear magnetic resonance; GC-TOF: Gas Chromatography Time-Of-Flight; HS–PTV: Headspace–Programmed Temperature Vaporization.

**Table 3 jcm-10-01723-t003:** Studies using alternative techniques.

Study	Population	Lung Cancer Patients (*n*)	Metabolites	Method/Device	Main Results
Takahashi Y., 2015 [31]	171	171	N^1^,N^12^-diacetylspermine	Colloid gold aggregation procedure	Sensitivity: 69.4% Specificity: 57.4% Accuracy: 60.8%
Takahashi Y., 2015 [32]	499	260	Diacetylspermine	Colloidal gold aggregation procedure	Sensitivity: 62.2% Specificity: 71.7%
Mazzone P.J., 2015 [33]	145	90	Volatile organic compounds analysis	Colorimetric sensor array	Sensitivity: 81.4% Specificity: 60.0%
Gào X., 2019 [34]	980	245	NO metabolites (nitrite and nitrate) 8-isoprostane	ELISA	
Gào X., 2018 [35]	866	207	8-isoprostane	ELISA	Accuracy: 62.4%
Zhang W., 2020 [36]	309	112	Ferritin light chain, Mitogen-Activated Protein Kinase 1 Interacting Protein 1 Like, Fibrinogen Beta Chain, Member RAS Oncogene Family RAB33B and RAB15	ELISA	Accuracy: 82.0–94.7%
Xia X., 2016 [37]	65	45	Midkine	ELISA	Sensitivity: 71.2% Specificity: 88.1%
Wang W., 2020 [36]	51	31	Kininogen 1 Osteopontin α-1-antitrypsin	ELISA	Sensitivity: 85–100% Specificity: 53–65%
Liu B., 2020 [38]	101	74	Gene: *CDO1*, *TAC1*, *HOXA*, *SOX17*	Methylation on beads and real-time PCR	Sensitivity: 93% Specificity: 30%
Nolen B.M., 2015 [39]	234	83	Insulin-like growth factor-binding protein 1, interleukin-1 receptor antagonist a, Carcinoembryonic antigen-related cell adhesion molecule 1	Multiplexed bead-based immunoassays	Sensitivity: 72% Specificity: 100% Accuracy: 71–83%
Wu Z., 2019 [40]	50	50	Cell-free DNA	Next-generation sequencing platform	Accuracy: 69%
Kawamoto H., 2019 [41]	178	54	Prostaglandin E-major urinary metabolite	Radioimmunoassay	Sensitivity: 67.7% Specificity: 70.4%

ELISA: enzyme-linked immunosorbent assay; PCR: Polymerase Chain Reaction.

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
