# Peer review of "Urinary Biomarkers for Early Diagnosis of Lung Cancer"

_jcm, 2021, doi:10.3390/jcm10081723_

Round 1
Reviewer 1 Report
I have no further concern and now paper should be accepted.
Author Response
Dear Reviewer,
We thank you for your kind replay.
The paper has been revised by a native English speaker.
Dott. Roberto Gasparri

Reviewer 2 Report
Perhaps as a result of extensive major reformatting the language has been compromised. I suggest an independent and careful language review. As for the content itself, I still think that the subject is extremely interesting. However, I find the reasoning followed by the authors sometimes confusing. II think it would benefit of some polishing.
Author Response
Dear Reviewer,
We have taken note of your valuable observations, for which we thank you.
The paper has been extensively revised for grammar and fluency by a native English Speaker.
Dott. Roberto Gasparri.

Reviewer 3 Report
The authors have successfully implemented all the suggested changes. The more profound ones regarding the objective and discussion as well.
Author Response
Dear Reviewer,
We thank you for your kind replay.
The paper has been revised by a native English speaker.
Dott. Roberto Gasparri

This manuscript is a resubmission of an earlier submission. The following is a list of the peer review reports and author responses from that submission.
Round 1
Reviewer 1 Report
The article reviewed by Roberto et al. regarding how urine can be used as a lung cancer biomarker source is an exciting topic. The authors nicely summarized the information. However, I have a few concerns, and that should be improved.
1) Please make some efforts to improve the abstract because the current version seems not elaborative.
2) The discussion part seems very unorganized and needs massive improvement. The reference paper should be written in a broader context, not always mentioned by et al.
3) The introduction part has somehow disconnected related to molecular biomarkers and disease findings.
4) The authors should provide some information about kidney biology
Reviewer 2 Report
Some English corrections, not an exhaustive list:
47. ..to pilot study revelling a great potentiality for early lung cancer diagnosis [13,14]. --> pilot studies revealing...
71. "On of older approach on urinary..." One of the oldest approaches...
72. Detection dogs are used currently to identify illegal elements (e.g. explosives, drugs) or recognize a missing person in highly demanding environments --> find
95. evaluate --> evaluated
123- Seven study [32–35,39–41] were immunosorbent assay --> plural. The whole manuscript must be revised for this type of error.
Reviewer 3 Report
First of all, I would like to thank the authors for the effort in the work presented. It is easy to read, but nevertheless, I would like to make a number of comments that would help to improve the work and better understand its overall objective.
The title should be modified according to the current state of the work presented since the first part of the title is adequate, but the second part does not describe the content of the paper. To the approach on urinary analysis employing canines in the detection of cancer, the authors use only two paragraphs, and artificial intelligence is simply named with a superficial analysis. However, these two concepts are fundamental to the title.
The abstract and introduction are correct, where is said that the paper aims to review and evaluate the researches on urine as a source of biomarkers for the early detection of lung cancer. But reading the paper, it is not easy to know whether the authors want to present a research paper, a meta-analysis of previous studies, a review on the current state of the subject, or their expert opinion on the trends in early diagnosis of lung cancer in the coming years.
In materials and methods, it is indicated that various combinations of database-specific controlled vocabulary were used, but these combinations and the Boolean searches employed are not specified. The number of articles found in each database is also not indicated, the total number of papers found by each of the two researchers, nor in how many papers the two authors disagree, nor in how many of them they disagree. This means that other authors will not be able to repeat the search and make this paper reproducible.
The sections on limitations and future perspectives do not make direct reference to any of the articles or groups of articles found in the search and only generically describe limitations by sample size. Also generically name approaches such as Big Data and artificial intelligence, but without providing any strategy or new idea in the field.
The summary is not really a summary of what is presented in the article, and the conclusions are a wish list of where the field should evolve but do not refer to or rely on the articles referenced in the results.